# Interventions during Early Infection: Opening a Window for an HIV Cure?

**DOI:** 10.3390/v16101588

**Published:** 2024-10-09

**Authors:** Christopher R. Hiner, April L. Mueller, Hang Su, Harris Goldstein

**Affiliations:** 1Department of Microbiology & Immunology, Albert Einstein College of Medicine, Bronx, NY 10461, USA; christopher.hiner@einsteinmed.edu (C.R.H.); april.mueller@einsteinmed.edu (A.L.M.); 2Department of Pediatrics, Albert Einstein College of Medicine, Bronx, NY 10461, USA

**Keywords:** early HIV intervention, HIV cure, early ART initiation, Block and Lock, Shock and Kill, broadly neutralizing antibody, adoptive CD8+ T cell transfer, gene therapy

## Abstract

Although combination antiretroviral therapy (ART) has been a landmark achievement for the treatment of human immunodeficiency virus (HIV), an HIV cure has remained elusive. Elimination of latent HIV reservoirs that persist throughout HIV infection is the most challenging barrier to an HIV cure. The progressive HIV infection is marked by the increasing size and diversity of latent HIV reservoirs until an effective immune response is mobilized, which can control but not eliminate HIV infection. The stalemate between HIV replication and the immune response is manifested by the establishment of a viral set point. ART initiation during the early stage limits HIV reservoir development, preserves immune function, improves the quality of life, and may lead to ART-free viral remission in a few people living with HIV (PLWH). However, for the overwhelming majority of PLWH, early ART initiation alone does not cure HIV, and lifelong ART is needed to sustain viral suppression. A critical area of research is focused on determining whether HIV could be functionally cured if additional treatments are provided alongside early ART. Several HIV interventions including Block and Lock, Shock and Kill, broadly neutralizing antibody (bNAb) therapy, adoptive CD8+ T cell therapy, and gene therapy have demonstrated delayed viral rebound and/or viral remission in animal models and/or some PLWH. Whether or not their application during early infection can improve the success of HIV remission is less studied. Herein, we review the current state of clinical and investigative HIV interventions and discuss their potential to improve the likelihood of post-treatment remission if initiated during early infection.

## 1. Introduction

In 1983, human immunodeficiency virus (HIV) was identified as the causative agent for acquired immune deficiency syndrome (AIDS) [1,2]. This infection was ultimately lethal until 1996, when combination antiretroviral therapy (ART) was shown to potently suppress HIV replication and became the standard of care. Currently, there are over 30 individual and combinational drug regimens available that target various stages of the HIV life cycle, halting viral replication. Effective ART treatment of people living with HIV (PLWH) preserves their immune function, prevents them from transmitting HIV, and provides them with a life expectancy comparable to the general population [3]. ART has transformed HIV from a fatal disease into a chronic disease requiring lifelong treatment. As a lentivirus, HIV integrates its genome into the host DNA and exploits the cellular machinery to produce viral genomic and structural RNA, facilitating HIV replication and the production of new virions to infect additional cells. Despite its potent capacity to completely suppress HIV replication and the infection of new cells, current ART does not eliminate infected cells containing integrated HIV DNA. Cells infected by HIV are cleared either by death of the infected cells after virus release or by effector immune cells through recognition and elimination of infected cells that present viral antigens. However, infected cells become invisible to the immune system when they stop producing viral proteins, enabling them to persist in lymphoid tissues for years. More importantly, a small fraction of these infected “sleeper” cells, which constitute latent HIV reservoirs, maintain their capacity to produce HIV after activation and thereby cause viral rebound when ART is discontinued. Studies have shown that HIV reservoirs are maintained through clonal expansion, as evidenced by the presence of identical proviral sequences in multiple infected cells within the same sample from ART-treated PLWH [4,5]. In addition, while only a small fraction of clonal cells may produce viral progeny and be cleared by immune surveillance, the majority remain silent and replenish the clone, making it nearly impossible to eradicate HIV reservoirs [6,7]. Consequently, this hinders the achievement of an HIV cure.

HIV was considered incurable until 2008, when Timothy Ray Brown, aka the “Berlin patient”, achieved an HIV cure after receiving two allogeneic hematopoietic stem cell transplants (allo-HSCTs) for acute myeloid leukemia [8]. The physicians employed an unprecedented strategy to cure his HIV infection by using a healthy bone marrow donor homozygous for the CCR5∆32 mutation. This enabled hematological reconstitution with donor-derived hematopoietic stem cells, which generated CD4+ T cells carrying the CCR5∆32 mutation, conferring resistance to R5-tropic HIV infection. This strategy successfully provided an HIV cure until the patient’s tragic death in 2020 from recurrent AML, marking a crucial milestone demonstrating that an HIV cure could be achieved. Similar therapeutic procedures have been performed in several other patients who displayed remission of HIV infection, thereby demonstrating that this is a valid approach for an HIV cure. Recently, a PLWH has been reported to remain in HIV remission for almost 3 years after receiving allo-HSCT from a CCR5 wild-type/wild-type (CCR5wt/wt) donor, becoming the first of its kind, expanding the potential application of allo-HSCT in the HIV cure field [9]. A recent cohort study recruited 30 PLWH (including the cured London patient [10] and Düsseldorf patient [11]) with hematological malignancies who received allo-HSCTs from donors either homozygous for the CCR5∆32 mutation or CCR5wt/wt [12]. Significantly accelerated HIV reservoir reduction was observed after full donor chimerism compared to before allo-HSCT, regardless of the donor CCR5 genotype, indicating that the transplantation process and donor immune cells are the major factors driving reservoir clearance [12]. However, the potential morbidity and mortality associated with allo-HSCT prevents expanded use of this strategy beyond PLWH with hematological malignancies who require allo-HSCT to treat their cancer. Additionally, the rarity of HLA-matched donors with CCR5∆32 mutation to provide the transplantation creates a further barrier to widespread implementation of allo-HSCT for an HIV cure.

A much less aggressive approach being pursued to achieve a “functional cure” rather than a “sterilizing cure” of HIV is exemplified by two groups of PLWH: elite controllers (ECs) and post-treatment controllers (PTCs) [13]. ECs (~1% of PLWH) can spontaneously maintain viral suppression in the absence of ART due to exceptional anti-HIV immune responses likely enabled by uniquely protective HLA alleles [14]. PTCs (~10% of PLWH), on the other hand, maintain viral suppression following ART cessation and only possess moderate HIV-specific CD8+ T cell activity [2], suggesting that ECs and PTCs accomplish HIV control through different mechanisms. It is noteworthy that both ECs and PTCs may lose viral control even after years of viral suppression without ART, suggesting dynamic interactions between HIV and the host immune system that keep virus under control, until this equilibrium is disturbed, enabling resurgence of active HIV replication.

Although the mechanisms of HIV control in PTCs are not fully understood, most identified PTCs initiated ART during the early stage of infection [15,16]. After HIV acquisition, there is an eclipse phase (~7–10 days) when neither viral RNA nor antiviral antibodies can be detected in the plasma, followed by acute HIV infection (~20 days), which can be diagnosed by detection of viral RNA but not of antiviral antibodies and is often associated with a flu-like acute retroviral syndrome. Early HIV infection extends to ~6 months after viral exposure, and ART initiation during this time frame is usually considered “early treatment” [17]. Early ART initiation has been shown to not only increase the opportunity among PLWH who display post-treatment viral control, but also to lower HIV set point, limit the establishment of HIV reservoirs, reduce the risk of HIV transmission, preserve immune function, reduce morbidity and mortality, and improve the overall life quality of PLWH as compared to PLWH who received delayed treatment [18,19,20,21,22,23]. Therefore, it is recommended that PLWH start ART treatment as soon as possible after HIV diagnosis [24]. However, early ART treatment still does not cure HIV, as viral reservoirs are established within days of viral exposure, before peripheral HIV can be detected [25,26,27]. Achieving an HIV cure likely requires additional interventions to prevent resurgence of HIV infection after ART cessation. Indeed, delayed viral rebound and/or HIV remission has been achieved in some animal models and PLWH who received a combination of ART and at least one other therapeutic intervention targeting the reservoirs [28,29,30,31,32,33,34,35]. While the majority of these interventions were administered to chronically infected ART-suppressed individuals who have received ART for at least 12 months, limited studies have tested their effects alongside ART during early infection when HIV sequences are less diversified and viral reservoirs are relatively unstable as compared to that during chronic infection [23,36]. In this review, we summarize major strategies for an HIV cure, including early ART initiation, Block and Lock, Shock and Kill, broadly neutralizing antibodies (bNAbs), adoptive CD8+ T cell transfer, and gene therapy, and discuss the rationale for their application as early HIV interventions that may promote the delivery of an HIV cure.

## 2. Early ART Initiation as an Early HIV Intervention?

Early ART initiation is so effective at limiting HIV spread that taking ART within 72 h post-HIV exposure is widely considered a valid means for preventing the establishment of viral infection [37]. When initiated beyond this period, while ART may be unable to fully prevent sustained HIV infection and generation of latent reservoirs, early ART treatment (usually considered within 6 months of HIV acquisition) can restrain the diversity and abundance of HIV reservoirs, which is associated with an improved duration of viral control after treatment interruption [38]. Therefore, early ART initiation is a major strategy to enable potential achievement of HIV remission exemplified by PTCs. The role of early ART intervention in reducing reservoir size, thereby enabling better viral control in PTCs compared to the larger reservoir size seen in non-controllers due to delayed treatment, is well established [15,16]. Notably, the realization of post-treatment control by early ART intervention is not exclusive to adult PLWH but also includes children and adolescent PLWH, which collectively constituted 6.4% of total PLWH and 20.8% of new infections in 2022 [39]. For instance, the Mississippi Baby initiated ART at 30 h after birth and maintained HIV remission for 27 months after treatment discontinuation [40]. A French teenager who also received ART after birth remained HIV-suppressed in the absence of treatment for 12 years as of 2015 [41]. In the recent IMPAACT P1115 cohort, 4 out of 6 children who started ART within 48 h of birth achieved HIV remission for at least 48 weeks during supervised analytical treatment interruption (ATI) [42]. An HIV cure might be more achievable in children than in adult PLWH through early ART intervention, as the reservoir size is much smaller in children due to the reduced number of target cells available for HIV infection at the time of ART initiation.

Another possible mechanism of HIV control exhibited by PTCs lies in the preservation of their immune function conferred by early ART initiation, which would otherwise be substantially compromised during untreated infection [43]. In SIV-infected macaques, early treatment supported the development of virus-specific memory CD8+ T cells that exhibited enhanced proliferative and antiviral cytotoxicity upon ART cessation, favoring delayed viral rebound [44]. In the VISCONTI cohort, PTCs displayed preserved polyfunctional HIV-specific CD4+ and CD8+ T cell responses [45]. Early ART initiation is especially beneficial to pediatric PLWH on protecting the compromised development of their immune system during early life [46,47]. The natural immune environment in infants is tolerogenic due to T helper 2 (Th2) and regulatory T cell-mediated anti-inflammatory and immunoregulatory responses that outweigh the Th1 cell-mediated inflammatory response [48]. As a result, HIV viremia and set point are significantly higher in ART-naïve pediatric PLWH than their adult counterparts [49], which accelerates progression to AIDS in the former group (1 year vs. 10 years) [48]. Early ART initiation rescues CD4 T cell counts and preserves polyfunctional antiviral T cell responses in addition to reducing the reservoir size in pediatric PLWH [50,51]. Together, preserved functional antiviral immune responses plus a limited number of HIV reservoirs provide pediatric PLWH with a higher probability of achieving an HIV cure than adult PLWH if receiving early treatment intervention.

### Future of Early ART Initiation

The inevitable rebound of HIV after ART discontinuation requires PLWH to stay on life-long ART, which can extend to greater than 60–70 years for pediatric and adolescent PLWH. Daily dosage with ART can be mentally and economically exhausting due to the challenges of strict adherence and potential social stigma preventing the implementation of early ART. Long-acting (LA) ART may overcome these challenges by reducing the frequency of ART administration from daily to monthly while maintaining its therapeutic effects [52,53,54]. At the time of this review, there are two LA regimens approved by the FDA [55]. Cabenuva consists of an integrase inhibitor cabotegravir (CAB) and a non-nucleoside reverse transcriptase inhibitor (NNRTI) rilpivirine (RPV). Several phase II and III trials have demonstrated that PLWH, including adolescents, receiving monthly or bimonthly intramuscular injection of Cabenuva achieved viral suppression at rates comparable to those on standard ART [56,57,58,59]. Lenacapavir (LEN) is a first-of-its-kind HIV capsid inhibitor. It can be administered twice a year in combination with other background regimens to maintain HIV suppression, especially in PLWH experiencing multidrug resistance [60,61]. Recently, a phase III trial demonstrated that the twice-yearly subcutaneous injection of LEN showed superior capacity for HIV prevention over conventional daily oral ART [62].

The convenience and high efficacy of LA ART has the potential to markedly improve HIV treatment and prevention. Moreover, individuals who received LA ART in trials expressed a preference of LA over conventional daily oral ART [57,63]. Recent studies also reported that LA ART improved adherence in PLWH who had difficulties maintaining daily oral ART [64,65]. The impact of LA ART initiated during early HIV infection remains to be tested. If LA ART proves as effective during early infection as it is during chronic infection, it may encourage PLWH to initiate ART early and remain on therapy. However, early ART intervention alone is unlikely to cure HIV. The addition of other interventions during early infection will likely be required to provide an opportunity to achieve HIV remission, as discussed below (Figure 1).

## 3. Reservoir Modulating Strategies

The “Block and Lock” and “Shock and Kill” strategies are two distinct approaches aimed at addressing one of the biggest challenges in curing HIV: the persistence of latent viral reservoirs despite ART. This section outlines how each of these strategies has the potential to be utilized in early treatment interventions.

### 3.1. Block and Lock

The “Block and Lock” strategy aims to prevent the reactivation of transcription of HIV provirus by forcing the integrated proviruses into a state of deep latency [66]. This is achieved using latency-promoting agents (LPAs) that silence the expression of the provirus by targeting either HIV- or host-specific factors to prevent viral transcription. An LPA with a “Block” phenotype inhibits HIV transcription, whereas an LPA with a “Lock” phenotype induces deep latent reservoirs to become resistant to reactivation, even in the absence of ART. For PLWH, the “Block and Lock” strategy proposes using LPAs alongside ART to suppress persistent viral transcription, reduce residual viremia, and prevent reseeding of the latent reservoirs. This combinational therapy may aid in the reduction in HIV latent reservoirs over time. Once viral recurrence is controlled, LPAs alone would continue to enforce deep and irreversible latency of HIV, achieving permanent silencing of proviruses without ART and ultimately leading to the elimination of HIV latent reservoirs. Below, we briefly describe LPAs being assessed for the “Block and Lock” strategy, and postulate how these treatments may be applied as an early intervention.

#### 3.1.1. Current State of the Block and Lock Strategy

Since the initial proposal of the “Block and Lock” strategy in 2017, with the description of the potent HIV Tat inhibitor didehydro-Cortistatin A (dCA) [67,68,69], several additional LPAs have been described, including small molecules that target HIV proteins [70,71,72], epigenetic modifiers [73,74,75,76,77,78,79,80], and cell signaling pathways [81,82,83,84,85,86]. Additional approaches have been reported for “Block and Lock” of latent HIV-infected cells using nucleic acids such as siRNA [87], long non-coding RNA [88,89], repurposed Clustered Regularly Interspaced Short Palindromic Repeats (CRISPR)/CRISPR Associated Protein 9 (Cas9) systems [90,91], oligonucleotide memetics [92], recombinant proteins [93,94], and nanosheets [95]. The mechanisms of action of these LPAs have been reviewed extensively elsewhere [96,97,98]. Notably, most of these studies have been conducted in the context of chronic HIV infection utilizing in vitro cell latency models, in vivo mouse models, or SIV-infected macaques. The function of LPAs was primarily assessed by evaluating either the duration until viral rebound following LPA treatment and ART treatment interruption or the resistance to latency reversal after combined treatment with LPAs and latency reversal agents (LRAs). Some LPAs have been assessed using ex vivo models derived from primary cells of PLWH; however, they have yet to be critically assessed for their efficacy as early intervention strategies.

#### 3.1.2. Block and Lock as an Early HIV Intervention?

It is generally accepted that the transcriptional activity of HIV provirus is influenced by its integration site and the surrounding chromatin environment [14,99,100]. Therefore, a promising “Block and Lock” strategy for early intervention in HIV may involve a “Lock” phenotype LPA that targets the formation of viral reservoirs by modifying initial integration of HIV into the genome during the early phase of infection. HIV provirus integrated into high-transcriptional genomic regions tends to have higher transcriptional activity, while provirus integrated into less active genomic regions exhibits lower transcriptional activity. Therefore, steering integration towards heterochromatin, regions with minimal transcriptional activity, may effectively silence the virus and limit initial viral spread. While not completely preventing integration, this approach may be an ideal alternative as it reduces the overall transcriptional activity of the virus, maintaining it in a deeply latent state and minimizing the risk of viral reactivation. Implementing such a strategy early in the course of infection could prevent the establishment of highly transcriptionally active viral reservoirs and potentially reduce the need for lifelong ART. For example, the small molecules referred to as LEDGINs show promise as early intervention LPAs due to their binding to the LEDGF/p75-binding pocket of the HIV integrase, which reduces integration, transcription, and reactivation [99,101,102,103]. The LEDGIN candidate GS-9822 was reported to redirect viral integration towards repressive chromatin regions, thereby reducing reactivation of the integrated provirus [101]. This study also showed through integration site sequencing that GS-9822 increases the proportion of latent provirus compared to productive provirus [101]. Unfortunately, GS-9822 was found to cause urothelial cytotoxicity in cynomolgus monkeys, and the development was halted. Nevertheless, this provided a proof-of-concept demonstration that LPAs that redirect HIV integration towards minimally active chromatin regions, such as LEDGINs, may offer a promising strategy for early intervention. By steering initial integration into these inactive regions, the virus remains transcriptionally inactive, which increases the likelihood of durable silencing after treatment discontinuation, thereby increasing the potential for achieving a functional cure. Furthermore, by reducing the size of productive viral reservoirs, this treatment might increase the effectiveness of other therapeutic approaches, such as genetic engineering or cellular therapies, to prevent the reactivation of infection from the reservoirs and providing a functional cure.

#### 3.1.3. Future of the Block and Lock Strategy

The “Block and Lock” strategy is still at its early stage of development and the clinical anti-HIV effects of LPAs remain to be determined. Future directions for “Block and Lock” strategies should increase the in vivo investigations on the identified compounds and continue to identify novel HIV transcription inhibitors [104]. In addition, while many current LPAs show promise on preventing HIV reactivation in the ex vivo studies and/or animal models, there are few studies which examine the effects of these compounds during the early phase of infection. Promising candidates, such as LEDGINs, need to be optimized to reduce toxicity while maintaining effectiveness, with studies utilizing in vivo models, including mouse and SIV studies. The results from these studies, aimed at refining these compounds to enhance safety and functional properties, could support their potential use in early treatment regimens to prevent the formation of extensive viral reservoirs and promote durable viral silencing after treatment discontinuation.

### 3.2. Shock and Kill

The “Shock and Kill” strategy, first coined in 2004 [105], is a two-stepped approach that aims to eradicate HIV from the body through reactivating and eliminating viral reservoirs. The first phase involves shocking viral reservoirs out of latency using LRAs, while the second phase focuses on killing the cells displaying newly active virus by either leveraging patient’s immune system or promoting apoptosis of infected cells [106]. Below, we briefly outline current research on LRAs used for the “Shock”, as well as the various “Kill” strategies, and postulate on how these treatments may be applied as an early intervention strategy.

#### 3.2.1. Current State of the Shock and Kill Strategy

Numerous classes of LRAs have been identified and evaluated, including protein kinase C agonists (PKCas) [107,108,109], histone deacetylase inhibitors (HDACis) [110,111,112,113], histone methyltransferase inhibitors (HMTis) [114], Bromodomain (BD) and Extra-Terminal Domain (ET) protein inhibitors [115], CCR5 antagonists [116], SMAC mimetics [117,118], and toll-like receptor (TLR) agonists [119,120,121]. The mechanisms of action of these compounds have been reviewed extensively elsewhere [105,122,123,124,125,126]. Additional new LRA candidates have recently been reported, including resveratrol analogs [127], the BCL-2 antagonist obatoclax [128], aptamers [129], 4-phenylquinoline-8-amine [130], and aminobisphosphonates [131], demonstrating the robust efforts to discover LRAs with new mechanisms. While clinical trials using LRAs alone or in combination with other therapeutics have reported increases in HIV production, none have shown a sufficient reduction in the size of viral reservoirs or a delay in viral rebound following the cessation of ART in PLWH [34,113,132,133,134,135,136,137,138,139,140,141,142]. This suggests that while LRAs can potently reactivate latent HIV reservoirs, they have not yet been effective in significantly decreasing the number of latently infected cells, either from insufficient “Shock” due to incomplete reactivation of the reservoirs, or from inadequate “Kill” due to the well-characterized exhaustion of the immune system of PLWH resulting in suboptimal killing of reactivated cells [143,144]. Because reactivating latently infected cells with LRAs alone is not sufficient to lead to their elimination, additional interventions are required to augment the “Kill” phase, such as therapeutic vaccines to enhance CD8+ T cell-mediated lysis [145,146,147], immune system-enhancing cytokines such as IL-15 [148,149,150,151], exhaustion-reversing checkpoint inhibitors like anti-PD-1 (which also acts as an LRA) [152,153,154], infusion of CD8+ T cells or NK cells [155,156], or implementing bNAbs (discussed later in this review). Although early in vitro and in vivo animal studies show promising results for these combined interventions, further investigation is needed to assess their effectiveness as treatments for PLWH. Additionally, “Shock and Kill” approaches have not yet been extensively evaluated as an early intervention strategy.

#### 3.2.2. Shock and Kill as an Early HIV Intervention?

The “Shock and Kill” strategy initiated within the first months after infection in combination with ART may be more effective due to the smaller and reduced heterogeneity of viral reservoirs at this stage and may prevent the formation of extensive viral reservoirs. Additionally, the relatively intact immune system, which is minimally compromised during early HIV infection, may provide a more potent “Kill” response which is capable of effectively targeting and eliminating reactivated latently infected cells. Indeed, a recent study tested this hypothesis in an HIV-infected humanized mouse model by co-administering an HDACi, suberoylanilide hydroxamic acid (SAHA) or panobinostat, alongside ART initiated ~1–2 months following HIV exposure [157]. It was hypothesized that this approach would prevent recently infected cells from entering latency, as shown in the pervious in vitro studies [158]. Interestingly, while cell-associated HIV RNA was suppressed to a similar level between the ART alone group and ART plus HDACi group, demonstrating the potent antiviral activity of ART on halting the viral life cycle, the cell-associated HIV DNA level was comparable or even higher in the ART plus HDACi group, possibly due to the less developed HIV-specific immune responses during early infection or compromised immune function induced by HDACi [159]. Consequently, although ART can efficiently inhibit active HIV replication, infected cells were not effectively eliminated after early intervention of ART plus HDACi, emphasizing the importance of enhancing the “Kill” in the “Shock and Kill” strategy to achieve an HIV cure.

Notably, a recent study demonstrated that LRAs can modulate HIV antigen processing and presentation, with HDACi and PKCa affecting the degradation patterns of HIV peptides in opposite ways: HDACis narrow the range of antigenic fragments, while PKCas broaden it [160]. Increasing the range of HIV peptide presentation by PKCa may generate a broader T cell response that may be useful during early infection, ensuring infected cells are readily recognized by a diverse population of CD8+ T cells. However, while this is an intriguing hypothesis, it has yet to be tested.

Ironically, some kill enhancement strategies may be detrimental in an early treatment strategy—in one study, six acutely SIV-infected rhesus macaques immediately treated with IL-15 showed an increase in viral set point despite higher levels of SIV-specific CD8+ T cells and NK cells [149]. As suggested by the authors, this may be due to IL-15-induced CD4+ T cell activation, which increased their susceptibility to infection [149]. In addition, it was also observed that the enhanced antiviral CD8+ T cells induced by N-803 can subsequently block the latency reversing activity of N-803 (143). Only after depleting CD8+ lymphocytes by treatment with a CD8 depleting antibody, N-803 administration induce robust and persistent HIV reactivation in vivo (143). The requirement for systemic depletion of CD8+ T cells may complicate the clinical application of N-803 as a latency reversing agent [161]. An alternative approach to generating a more potent “Kill” response is the design of therapeutic vaccines. Promising results from one therapeutic vaccine, administered to PLWH who received early ART intervention (~6 months from viral acquisition), showed a shift in the predominant targets of HIV-specific T cells toward conserved regions of the virus, indicating the plasticity of the immune system during an early stage of infection [162]. When initiated as an early intervention, therapeutic vaccination may exploit a less diverse viral genome and an intact immune response.

#### 3.2.3. Future of the Shock and Kill Strategy

Future research into LRAs should consider including them as part of an early intervention strategy in combination with ART, which would capitalize on the smaller and less heterogeneous viral reservoirs present during early infection. The addition of immune enhancers could facilitate targeting and eliminating reactivated virus induced by LRAs, leveraging the robust antiviral function of CD8+ T cells and NK cells. A greater understanding of the effects of LRAs on HIV antigen processing and presentation, their impact on CD8+ T cell responses, and the role of CD8+ T cells in latency reversal may enable researchers to devise methods to modulate LRAs to display latency reversing activity while maintaining the antiviral activity of CD8+ T cells and develop an effective “Shock and Kill” therapy. Finally, understanding the potential benefits and drawbacks of different “Kill” strategies, including both CD8+ T cell-dependent and -independent modalities such as antibodies or NK cells, should help design more effective early interventions. Successful treatment strategies will likely require a combination of therapies, including efficient reactivation of the small number of latent proviruses in early infection and also targeted immunological enhancement to amplify the anti-HIV immune response. These efforts are essential for developing more effective interventions and achieving durable viral suppression, and possibly a cure for HIV.

## 4. Broadly Neutralizing Antibodies

Antibodies are a powerful multifunctional component of the adaptive immune response. Of particular interest for viral infections, neutralizing antibodies are a subgroup of antibodies that can prevent infection of host cells by blocking viral entry. Through interactions between the antibody Fc domain and FcγRs on immune effector cells, some of these antibodies can also facilitate Fc-mediated effector functions to eliminate infected cells. Recently, a subclass of anti-HIV neutralizing antibodies known as broadly neutralizing antibodies (bNAbs) has garnered significant interest for their potent capacity to treat and prevent HIV infection [163]. Through the process of somatic hypermutation (SHM) and affinity maturation, these bNAbs co-evolve with HIV to target conserved regions of the highly diverse HIV Env, decreasing the capacity of HIV to become resistant through immune escape mutations [164]. Several categories of bNAbs that bind various components of the HIV Env trimer, including the CD4 binding site, V1V2 loop, V3-glycan, silent face, gp120–gp41 subunit interface, fusion peptide, and the membrane proximal external region (MPER), have been previously reviewed [165,166,167]. Some proposed approaches for HIV treatment using bNAbs include passive transfer of bNAbs, genetic engineering of B cells or other cells to induce secretion of defined bNAbs, and vaccination with germline-targeting immunogens to guide development of endogenous bNAbs [168,169]. Below, we review recent advancements in these approaches and discuss the appealing features of bNAb-based therapies as early interventions targeting HIV infection.

### 4.1. Current State of bNAb Therapy

Passive transfer of bNAbs or engineered bNAb-derivatives, either as monoclonal antibodies or cocktails, is an effective HIV treatment that has been described and reviewed extensively elsewhere [165,167,170,171]. Numerous pre-clinical studies have been performed using animal models to assess the treatment efficacy of bNAb therapy for control of HIV post-infection [165]. Early studies found that the administration of a mix of bNAbs recognizing different epitopes suppressed viremia to undetectable levels in humanized mice and the virus was unable to escape immune pressure by envelope mutations [172]. Investigation of bNAb treatment of chronic SHIV infection demonstrated measurable levels of viral control in several studies, but viral rebound almost always occurred once serum bNAb levels dropped below therapeutic levels following treatment cessation [173,174,175,176]. Several studies of bNAb activity in humanized mouse models demonstrated that, in addition to their neutralizing activity, their capacity to recruit Fc-mediated effector functions to clear infected cells plays an important role in controlling [177,178,179,180]. The additional role for non-neutralizing functions of HIV-specific bNAbs for their activity was confirmed by studies in SHIV-infected rhesus macaques demonstrating that bNAb treatment with intact Fc-mediated functions led to a sharper decline in plasma viral load and cell-associated SHIV RNA in PBMC than treatment with Fc-null mutant bNAbs [173,181]. Assessment of bNAb treatment for chronic infection in PLWH in clinical trials has largely recapitulated the results of bNAb therapy of chronic SHIV infection in non-human primates. Initiation of bNAb therapy (3BNC117 and/or 10-1074) during ATI in PLWH demonstrated measurable reduction in plasma viral load, marked activation of HIV-specific CD8+ T cells, and a reduction in the intact viral reservoirs, but viral rebound was observed in nearly all treated individuals when serum bNAb levels decreased below therapeutic levels [31,32,182,183,184]. While these studies demonstrated that passive bNAb therapy suppressed HIV infection concurrent with the maintenance of circulating bNAb levels, they did not demonstrate the capacity of bNAb infusion to provide universal post-treatment control of chronic infection.

### 4.2. Therapy with bNAb as an Early HIV Intervention?

In addition to studies investigating the impact of bNAb treatment on the post-treatment control during the chronic stage of HIV infection, other studies have evaluated whether bNAb administration as an early HIV intervention could enable viral remission in the absence of ART. A study in macaques demonstrated that bNAb therapy initiated 3 days after SHIV infection enabled the emergence of a potent CD8+ T-cell immune response capable of providing sustained suppression of virus replication in 6 of 13 bNAb-treated monkeys [185]. Similar results were reported in a clinical study where 3BNC117 bNAb therapy was administered 1 week following ART initiation in PLWH; ART-free virologic control was observed following ATI in greater than 75% of treated individuals who harbored the 3BNC117-sensitive virus with viral control associated with enhanced anti-HIV CD8+ T cell responses [34,186]. One individual in this study had maintained ART-free control of HIV viremia for >4 years following a combination treatment of ART + 3BNC117 + the LRA romidepsin [186]. Notably, these studies highlight a potential vaccinal effect of bNAb therapy on CD8+ T cell activity by activating antigen-presenting cells through the formation of antibody:HIV-1-antigen immune complex, underscoring the benefits of bNAb administration in the first weeks after diagnosis when the immune response may not yet be compromised by HIV infection [187]. These findings suggest a potential role for bNAb therapy initiated alongside ART to induce post-treatment ART-free HIV control and support further studies to determine the long-term benefits of early bNAb treatment on the clinical course of HIV infection and the potential ability of longer duration bNAb treatment to induce universal ART-free remission.

A limitation of bNAb therapy is the need for repeated infusions to maintain therapeutic levels. One approach to overcome this limitation is the administration of adeno-associated virus (AAV) vectors encoding bNAbs, which provided sustained serum bNAb levels in humanized mice and extended viral control following termination of passive bNAb therapy [188]. An alternative strategy, which effectively produced sustained neutralizing titers of bNAbs in mice, is to molecularly engineer B cells to express bNAbs using clustered regularly interspaced short palindromic repeats (CRISPR)/Cas-based editing [189,190,191,192,193]. However, more research is needed to ensure the safety and efficacy of these approaches. Another encouraging strategy to provide sustained bNAb levels is by germline-targeting vaccination, initially designed for HIV prevention [194,195], and subsequently also proposed for therapy of PLWH [196]. Germline-targeting vaccination primes germline B cells that express bNAb precursor antibodies followed by boosts with engineered immunogens to drive somatic hypermutation (SHM) into the development of bNAbs [197]. Although germline-targeting vaccination strategies are still in the early stages of development, a first-in-human trial demonstrated that a priming vaccine could activate B cell precursors for a VRC01-class bNAb [198]. Recent advancements in the design of lipid nanoparticle (LNP)-encapsulated nucleoside mRNA (mRNA-LNP) vaccines, which enabled the resolution of the SARS-CoV-2 pandemic, have been leveraged to construct germline-targeting HIV vaccines that effectively induced affinity maturation of bNAb precursors to generate bNAb-like antibodies, suggesting that mRNA-LNP vaccines could be used to generate bNAbs to enable sustained control of HIV in PLWH [199,200,201]. Identification of optimal immunogens and modalities of vaccination may enable the development of vaccines capable of generating bNAbs that could be administered as an early intervention strategy for HIV treatment while the individual is ART-suppressed. This strategy could leverage the previously discussed advantages of early ART initiation while simultaneously generating sustained production of bNAbs and anti-HIV CD8+ T cell responses to provide durable treatment-free control of viremia.

### 4.3. Future of bNAb Therapy

Future research into bNAb therapy as an HIV intervention should assess the advantages of initiating treatment in the early stages of infection, particularly in the first months after exposure. Evidence of the direct effects of passive bNAb therapy on viral suppression and their capacity to act as a vaccine to induce anti-HIV CD8+ T cell responses described above are extremely encouraging. Furthermore, clinical trials have demonstrated the safety and feasibility of bNAb treatment for wider implementation. Although future studies will be required to determine the durability of bNAb-based therapies for achieving a treatment-free HIV cure, advancements in bNAb vaccine and transgenic bNAb-secreting cell technologies could improve clinical outcomes.

## 5. Adoptive CD8+ T Cell Therapy

Many proposed strategies for the cure of HIV rely on the unique abilities of adaptive immune effectors to eliminate or control viral infection. Substantial interest has been given to adoptive cell therapy of CD8+ cytotoxic lymphocytes (CTLs) [156,202,203]. The role of native CD8+ T cells in the control of HIV has been studied extensively and has been comprehensively reviewed elsewhere [204,205,206]. Through recognition of MHC class I-presented HIV-derived antigenic epitopes by their T cell receptors (TCRs), CTLs can directly target and eliminate HIV-infected cells and control HIV infection. However, the natural response of these cells is often compromised by exhaustion and immune escape by HIV that compromise their ability to achieve treatment-free control or sterilization of infection. As a result, reinforcement of endogenous anti-HIV T cell responses with adoptive cell transfer of ex vivo modified T cells is an attractive strategy to overcome these challenges. Several approaches have been proposed for generating HIV-specific CD8+ T cells with enhanced anti-HIV activity for adoptive cell transfer, including ex vivo expansion of autologous HIV-specific CD8+ T cells, transgenic expression of favorable HIV-specific TCRs, and expression of HIV-specific artificial chimeric antigen receptors (CARs) [156]. Below we briefly review adoptive CD8+ T cell therapy for HIV and assess its potential as an early intervention.

### 5.1. Current State of CD8+ T Cell Therapy for HIV

Early studies investigating the efficacy of adoptive cell therapy for HIV used HIV-specific CD8+ T cells expanded by ex vivo peptide stimulation to amplify the HIV-specific CD8+ T cell response [207,208,209]. While these studies indicated some increase in the antiviral activity of reinfused CD8+ T cells, long-term engraftment of the infused cells, which is crucial for durable HIV control, was not observed. Subsequent studies investigating the use of CD8+ T cells molecularly engineered to express defined HIV-specific TCR (TCR T cells) demonstrated that TCR T cells specific for the immunodominant Gag-derived epitope, SL9, potently inhibited HIV infection both in vitro and in vivo [210,211]. Although adoptive cell therapy of anti-HIV TCR T cells remains promising, clinical trials are needed to validate their safety and efficacy.

Another strategy for adoptive CD8+ T cell therapy for HIV has focused on using HIV-specific CAR-T cells that target HIV Env on the surface of HIV-infected cells in an MHC-unrestricted manner [212]. This approach is supported by in vitro studies indicating that these CAR-T cells provide superior control of HIV infection when compared to T cells expressing HIV-specific TCR [213]. An overview of the use of CAR-T cells for treating HIV has been provided elsewhere [214,215,216]. The use of CAR-T cells to treat HIV infection has been investigated since the 1990s, with early demonstration of specific lysis of HIV-infected cells by CAR-T cells followed by clinical trials, which demonstrated the safety and feasibility of this therapy but failed to demonstrate control of viral rebound or depletion of HIV reservoirs [217,218,219,220,221]. These early clinical trials utilized first-generation CAR-T cells consisting of a CD4-based CAR for binding of HIV Env and the intracellular CD3ζ subunit for signal transduction. More recent improvements of CAR design, which include intracellular costimulatory domains to promote activity and persistence, have greatly increased their functional activity compared to first-generation CAR-T cells, as indicated by their significant success in the treatment of cancer [222]. Application of these design advancements has reinvigorated interest in utilizing the CAR-T cell strategy for the treatment of HIV infection. Further advancements are being integrated to improve the function of CAR-T cells for HIV treatment, including bNAb-derived scFvs for higher-affinity Env binding and infection resistance [223,224,225,226,227,228], CAR-T cell-intrinsic immune checkpoint inhibition [227,229,230], and transduction of stem cells to generate CAR-T cells with favorable self-renewal, trafficking, and persistence phenotypes [231,232]. While the safety and efficacy of some of these CAR-T cell strategies are being evaluated in clinical trials for PLWH with established infection, the efficacy of these advancements to enable CAR-T cells to mediate ART-free remission when administered soon after infection has yet to be evaluated.

### 5.2. Adoptive CD8+ T Cell Therapy as an Early HIV Intervention?

Adoptive CD8+ T cell therapy has predominantly been administered to chronically infected PLWH to enhance their capacity to control HIV infection. Nevertheless, several features of this therapeutic modality make it an attractive early intervention for HIV. First, potent CD8+ T cell activity is thought to be an important correlate for ART-free control of HIV infection. Furthermore, the effectiveness of adoptive T cell therapy initiated within the first weeks to months after HIV exposure may be improved because of a reduced likelihood that T cell-resistant HIV variants would be present at the onset of treatment. Additionally, in combination with the “Shock and Kill” approach, early adoption of CD8+ T cell therapy could enhance the elimination of latent reservoirs that are smaller and less diverse during early infection. Importantly, cell therapies can also provide durable protection due to their behavior as a “living drug”, with long-term persistence becoming feasible with the latest advancements in cell engineering. To this end, initiation of adoptive CD8+ T cell therapy in the context of early HIV infection may provide better outcomes for viral control as compared to its administration during chronic infection.

### 5.3. Future of Adoptive CD8+ T Cell Therapy

Future studies should consider a role for early initiation of adoptive CD8+ T cell therapies, particularly as part of a multi-faceted strategy in combination with other treatment modalities, including ART, LRAs, and/or bNAbs. CD8+ T cell therapies could utilize HIV-specific CAR-T cells, endogenous HIV-specific T cells expanded by ex vivo stimulation, and T cells engineered to express HIV-specific TCR. Other strategies to improve the in vivo function and persistence of adoptively transferred CD8+ T cells, such as transduction of stem-like transcription factor TCF-1, should be explored [233]. With future optimization of cell products and timing of intervention, adoptive CD8+ T cell therapy could be a cornerstone of HIV cure strategies.

## 6. Targeted Gene Editing of HIV Provirus or Host Factors

Gene editing therapy for HIV aims to directly target and alter HIV provirus in infected cells or disrupt essential host factors required by HIV infection for productive viral replication to prevent infection. In doing so, these approaches could provide a means to deplete HIV proviral reservoirs or prevent the dissemination of the virus within PLWH, particularly after the cessation of ART. This can be achieved through several modalities that introduce targeted double-strand breaks at the locus of interest, which is then followed by nonhomologous end joining (NHEJ) for gene disruption or homology-directed repair (HDR) for precise gene editing. Several nucleases have been utilized for this purpose, including zinc finger nucleases (ZFNs), transcription activator-like effector nucleases (TALENs), and CRISPR/Cas9 systems, as reviewed elsewhere [234,235,236,237,238,239]. In this section we review the existing and proposed strategies for the use of gene editing to prevent viral entry through disruption of essential coreceptors or to inactivate HIV provirus, and propose how such strategies could be utilized for an early intervention to cure HIV.

### 6.1. Current State of Gene Editing Therapy for HIV

One of the earliest proposed targets for gene editing to treat or cure HIV infection was CCR5, a nonessential G-protein coupled chemokine receptor, which is also the most common co-receptor for entry used by HIV after binding CD4 [240]. Further support for targeting the CCR5 gene was provided by observation that its disruption by a naturally occurring 32 bp deletion (CCR5Δ32) confers protection from HIV infection in homozygotes for the mutated gene and delays disease progression in heterozygotes for the mutated gene [241,242,243]. The feasibility of achieving a generalizable HIV cure in PLWH through repopulation with CCR5-disrupted CD4+ T cells was indicated by the first demonstration of HIV cure in the “Berlin patient”, a PLWH who became HIV-free after allo-HSCT from a homozygous CCR5Δ32 donor [8,244]. This report was followed by considerable ongoing research to use gene editing technologies to provide this curative strategy to PLWH. Advancements in utilizing ZFNs [245,246,247,248,249], TALENs [250,251], and CRISPR [252,253,254,255] for editing of the CCR5 gene have demonstrated that effective ablation of CCR5 expression in CD4+ T cells or hematopoietic stem and progenitor cells effectively protect these cells and their progeny from infection by CCR5-tropic (R5-tropic) HIV virus. Of note, a recent clinical trial provided evidence that infusion of ex vivo CCR5-edited CD4+ T cells could delay viral rebound and that post-rebound control of infection was possible in 1 of 9 CCR5wt/wt PLWH [249]. Although most research into genome editing of coreceptors for HIV has focused on CCR5, there has also been development of genome editing strategies for CXCR4, the major co-receptor for CXCR4-tropic (X4-tropic) HIV strains. The CRISPR-Cas9 system has proven effective in editing CXCR4 and protecting these cells from infection by X4-tropic HIV strains [256,257]. However, ablation of CXCR4 may only be possible in mature CD4+ T cells because CXCR4 editing could adversely affect hematopoiesis and thymic differentiation due to the crucial role of CXCR4 in hematopoiesis and the subsequent migration and homing of progenitor cells [258,259]. Although infusion of genome-edited co-receptors in HIV permissive cells has demonstrated some promising results with early evidence of clinical safety, more research is needed to increase the efficiency of gene ablation in target cells and determine the subsequent impact of HIV infection of these interventions.

Another promising application for the CRISPR/Cas system is to target and disrupt the integrated HIV provirus, the determinant of HIV infection that is the core of latent reservoirs. Directly targeting HIV provirus by gene editing could irreparably inactivate the virus life cycle, deplete the reservoir, and potentially lead to a cure. Significant advancements have been made that have supported the feasibility of this strategy [260]. In these approaches, CRISPR guide RNAs (gRNA) are designed to target the HIV genome and utilize Cas-based cleavage to delete essential genes and regulatory elements. Several studies show that infected cells targeted for editing resulted in significant suppression of virus production [261,262,263,264,265]. A recent study in humanized mice found that LA ART paired with a CRISPR system targeting CCR5 and/or HIV provirus could effectively reduce the viral setpoint and the incidence of viral rebound [266]. Although these genome editing strategies have primarily been considered for providing a cure to PLWH during the chronic phase of infection, genome editing could be even more effective in curing HIV when administered as an early intervention.

### 6.2. Gene Editing Therapy as an Early HIV Intervention?

R5-tropic viruses are most likely to initiate infection and are the predominant HIV population during the early stages of infection, and it often takes years to observe a shift to X4 tropism [240,267]. Genome editing of CCR5 during the first months of infection may prevent or delay infection of new cells and achieve treatment-free control of infection without the need for targeting CXCR4 or other host factors with genome editing. Recapitulating the protective CCR5Δ32 phenotype in CCR5wt/wt individuals using CCR5-targeted gene disruption during early infection is more achievable than during chronic infection.

One of the major challenges of an HIV provirus-targeting strategy is the considerable genetic diversity of HIV within viral reservoirs, which may enable viral escape from CRISPR-based genome editing approaches [268]. Therefore, applying gene editing before the chronic stage of HIV infection, when viral sequences are less diverse, may help circumvent this issue. Furthermore, recent multiplexed approaches that simultaneously target several components of HIV provirus should reduce the impact of genetic diversity on the efficacy of CRISPR-based gene editing. Future clinical trials that study the outcomes of targeting HIV provirus with CRISPR-based editing should consider the benefits of investigating the efficacy of this approach as an early intervention.

### 6.3. Future of Gene Editing Therapy for HIV

Although immensely promising, several concerns need to be considered before wider use of genome editing approaches for HIV treatment, particularly the possibility of off-target DNA damage induced by nucleases. This safety concern has led researchers to develop sequencing technologies to surveille DNA cleavage events induced by off-target Cas activity with the intention to develop safer therapeutics [269,270]. In addition, vector delivery of the CRISPR complex may be unable to target all the latent viral compartments including the brain and lymphoid tissues, and consequently leave persisting viral reservoirs sufficient to mediate viral rebound after ART cessation. Considerable work is still needed to implement gene editing techniques as an early intervention for an HIV cure, particularly to optimize safety, delivery methods, and in vivo efficacy of these approaches. Nonetheless, gene editing remains an exciting approach and should be considered a priority for future development.

## 7. Conclusions

Despite remaining extremely difficult, achieving a sterilizing or functional HIV cure is the ultimate goal for HIV research [271,272,273]. While PTCs demonstrate the possibility of ART-free HIV remission, the majority of PLWH experience viral rebound upon treatment interruption, even with early ART initiation. Even so, there may still be a window within the first months of infection when advanced treatment interventions are able to cure HIV. Although early ART initiation does not completely prevent the establishment of HIV latency, it limits the initial reservoir size and induces a rapid decline in viral reservoirs, setting the foundation for HIV clearance by additional interventions. Indeed, individual or combined therapeutic approaches—including Shock and Kill, Block and Lock, bNAbs, adoptive CD8+ T cell transfer, and gene therapy—have delivered HIV remission in a subset of animal models and/or PLWH when administered during chronic infection. Incorporating these approaches with ART during the early stages of HIV infection may further deplete the smaller and less established viral reservoirs and thereby open a window for an HIV cure.

## Figures and Tables

**Figure 1 viruses-16-01588-f001:**
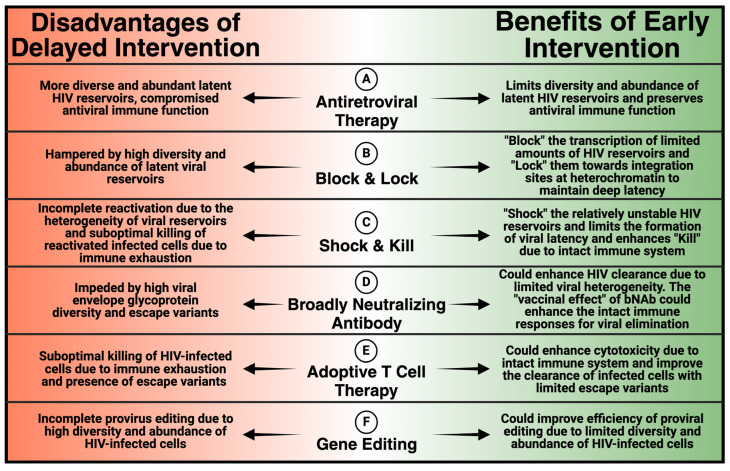
Benefits of early intervensions that may facilitate the achievement of an HIV cure. (**A**) Antiretroviral Therapy (ART): Early ART initiation can effectively limit the diversity and abundance of latent HIV reservoirs that are the major obstacle to an HIV cure. In addition, early ART intervention can preserve HIV-specific immune responses, which otherwise would be compromised during delayed treatment. (**B**) Block and Lock: An early “Block and Lock” strategy may take advantage of the smaller reservoir size and more effectively suppress HIV transcription, prevent viral dissemination, and limit formation of viral reservoirs. Additionally, certain latency-promoting agents (LPAs) can redirect viral integration towards repressive chromatin regions, inducing a state of deep latency that is resistant to reactivation, leading to ART-free HIV remission. (**C**) Shock and Kill: Early “Shock and Kill” intervations may limit the formation of latent reservoirs, and exhibit enhanced killing of infected cells with intact immune system. (**D**) Broadly Neutralizing Antibody (bNAb): Early bNAb therapy may benefit from the lower viral diversity and display enhanced HIV clearance through direct viral neutralization and Fc-mediated antibody-dependent cellular cytotoxicity (ADCC). In addition, the antibody–antigen complex displays a vaccinal effect that mobilizes HIV-specific CD8+ T cell activity, which is preserved during early infection to facilitate viral clearance. (**E**) Adoptive CD8+ T cell Therapy: Early adoptive CD8+ T cell therapy may display enhanced cytotoxicity due to the intact immune system, and effectively clear HIV-infected cells with limited presence of escape variants. (**F**) Gene Therapy: Early gene therapy may display more efficient proviral genomic editing due to lower viral diversity and smaller amount of infected cells, limiting the size of HIV reservoirs and increasing the opportunity of achieving ART-free viral remission.

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
