# Peer review of "Interventions during Early Infection: Opening a Window for an HIV Cure?"

_viruses, 2024, doi:10.3390/v16101588_

Round 1

Reviewer 1 Report

Comments and Suggestions for Authors

This review describes the potential interventions, based on the available literature, in order to achieve a functional cure rather a sterilization ( such as with HSV ). However,  the incorporation of the described interventions along with the early initiation of HAART remains a theoretical scenario and the impact for a functional cure remains unknown and much more efforts are needed

Author Response

Comments: This review describes the potential interventions, based on the available literature, in order to achieve a functional cure rather a sterilization ( such as with HSV ). However,  the incorporation of the described interventions along with the early initiation of HAART remains a theoretical scenario and the impact for a functional cure remains unknown and much more efforts are needed.

Response: Thank you for pointing this out. We acknowledge the paucity of publications that test the combination of early HAART plus other interventions on an HIV cure. And this is exactly why we want to bring this idea to the field based on previous literature and rational speculations. There is no doubt that early ART initiation alone can effectively limit reservoir size if not completely prevent HIV relapse post treatment interruption. Novel therapeutics have recently been developed and evaluated which showed potent antiviral function and led to HIV remission in a subset of animal models/PLWH, especially when combined with early ART. However, a majority of studies administered these therapeutic modules during chronic infection. It is reasonable to speculate that improved outcomes could be achieved if these novel therapeutics would be administered during early infection when HIV reservoirs are less abundant and diverse while immune function is preserved. Indeed, recently our colleagues have started to test these ideas including Dr. Garcia’s group (Lijun Ling et al, mBio, 2024) that administered HDAC inhibitors to hu-BLT mice as ART initiation, as well as Dr. Ole S. Søgaard’s group (Miriam Rosás-Umbert et al, Nature Communications, 2022; Jesper D. Gunst et al, Nature Medicine, 2022) that administered 3BNC117 bNAb to PLWH 1 week following ART initiation. These pioneering studies already showed some promising results on long-term HIV control which we believe should encourage more colleagues to investigate early implementation of novel treatment modalities. Here we aim to emphasize these pioneering studies and share our opinions with colleagues. Indeed, we are not the first to bring up these ideas but motivated by our colleagues (Stephen J Kent et al, Lancet Infectious Dieases, 2013; Nicolas Chomont et al, AIDS, 2018). Here we aim to reinforce these ideas and work together with our colleagues to test these ideas in order to potentially provide a pathway to the functional cure of HIV.

Reviewer 2 Report

Comments and Suggestions for Authors

This review of HIV cure is comprehensive and should be well received by the journal's readership.  I do not have any edits for the authors to consider.  I congratulate the authors on this comprehensive review. 

Author Response

Comments: This review of HIV cure is comprehensive and should be well received by the journal's readership.  I do not have any edits for the authors to consider.  I congratulate the authors on this comprehensive review.

Response: We sincerely appreciate the reviewer’s support and approval of our manuscript.

Reviewer 3 Report

Comments and Suggestions for Authors

The manuscript by Hiner et al reviews the advantage of intervention during early HIV infection which can lead to a positive outcome for the infected individuals opening a possible window for HIV cure.  The review uses basis of the finding that early ART initiation is so effective at limiting HIV spread that taking ART within 72 hours post-HIV exposure is widely considered a valid means for preventing the establishment of viral infection.  Within this window of time treatment an improved duration viral control with restrained diversity and abundance of HIV reservoirs can result after treatment interruption. In the context of early intervention, the review discusses the different possible treatments that include ART as well as other possible future approaches such as block and lock, shock and kill, broadly neutralizing antibodies, adaptive CD8+ T cell therapy and targeted gene editing.  The contents of the manuscript have previously been reviewed by several times by others. The current review is mainly a rehash of previous reviews with the assertion that early intervention could help improve duration of viral control.  The following points could significantly enhance the manuscript.

1.      As the authors state the possible window for their main point is within the first 72 hours.  The concern is whether infected individuals within 72 hours of infection will have symptoms to seek treatment.  Is there a specific population target for such approach?  The authors should expand on such issue and direct the content of the review to this point instead of just rehashing previously reviewed topics.

2.     The author describes gene editing as a potential treatment early intervention. What is the likely of gene editing approach can be ready for treatment within 72 hours window time?  The same concern could also apply for CD8+ T cells therapy.  Would the time it takes to expands these cells allow the 72 hours window of opportunity for early intervention? Addressing these concerns and expanding the review with inclusion of such points could enhance the reviews. 

3.     Lines 149-154.  ……. “Early ART initiation is especially beneficial to pediatric PLWH….. probability of achieving an HIV cure than adult PLWH if receiving early treatment intervention.” Should be expanded and clarified so that readers of the review can clearly understand.

Author Response

Comment 1: The manuscript by Hiner et al reviews the advantage of intervention during early HIV infection which can lead to a positive outcome for the infected individuals opening a possible window for HIV cure.  The review uses basis of the finding that early ART initiation is so effective at limiting HIV spread that taking ART within 72 hours post-HIV exposure is widely considered a valid means for preventing the establishment of viral infection.  Within this window of time treatment an improved duration viral control with restrained diversity and abundance of HIV reservoirs can result after treatment interruption. In the context of early intervention, the review discusses the different possible treatments that include ART as well as other possible future approaches such as block and lock, shock and kill, broadly neutralizing antibodies, adaptive CD8+ T cell therapy and targeted gene editing.  The contents of the manuscript have previously been reviewed by several times by others. The current review is mainly a rehash of previous reviews with the assertion that early intervention could help improve duration of viral control.  The following points could significantly enhance the manuscript.

Response 1: Thank you for your comments. We totally agree with the reviewer that the benefits of early ART as well as general HIV cure strategies have been reviewed exhaustively. Therefore, in the current manuscript, we briefly summarized these discoveries to set up the stage for our major focus: the timing. By far, in the majority of cure studies, these therapeutic modalities (block and lock, shock and kill, broadly neutralizing antibodies, adaptive CD8+ T cell therapy and targeted gene editing) were administered after a long gap post ART initiation (>12 months). Even though ART might be initiated during early HIV infection (~6 months of HIV acquisition), these therapeutic modules were initiated during chronic infection when reservoirs are more stable and immune function is impaired. Therefore, here we propose to prepone the initiation of additional interventions not long (within weeks) after early ART initiation to potentially improve the success of long-term HIV control, given the relatively unstable, less abundant and diverse reservoirs, as well as more preserved antiviral immune function during the early stage of infection. As in our response to Reviewer 1, we are fully aware that there have been very few studies testing these ideas but our colleagues have performed some pioneering studies and made promising progress. We would like to emphasize these studies and share our opinions building on these observations in the current manuscript to draw attention from more colleagues to investigate. This is the angle of our manuscript that differentiates it from previous literature.

Comment 2: As the authors state the possible window for their main point is within the first 72 hours. The concern is whether infected individuals within 72 hours of infection will have symptoms to seek treatment.  Is there a specific population target for such approach?  The authors should expand on such issue and direct the content of the review to this point instead of just rehashing previously reviewed topics.

Response 2: We apologize for the ambiguity, but our focus is extended to early HIV infection (~6 months after HIV acquisition). As beyond 72 hours, ART alone is not able to lead to an HIV cure, we propose to employ additional therapeutic modules during the early stage of infection (~6 months after HIV acquisition) which may enhance HIV killing and possibly open a window for an HIV cure. To test our ideas, we believe the field should consider performing studies where PLWH initiate ART during acute infection (~4 weeks after HIV acquisition) and then receive additional therapeutics within weeks of ART initiation to evaluate if early combinational therapy can lead to a better post-treatment control of HIV as compared to ART alone. If it does, we propose further investigation if a similar result could be achieved when combination therapy is initiated at a later point, such as 2+ months after HIV acquisition.

We have altered descriptions of timing in several parts of the revised manuscript in red text to enhance clarity.

Comment 3: The author describes gene editing as a potential treatment early intervention. What is the likely of gene editing approach can be ready for treatment within 72 hours window time? The same concern could also apply for CD8+ T cells therapy.  Would the time it takes to expands these cells allow the 72 hours window of opportunity for early intervention? Addressing these concerns and expanding the review with inclusion of such points could enhance the reviews.

Response: We apologize again for the ambiguity of timing. These issues have been addressed in the revised manuscript as labeled in red.

Comment 4: Lines 149-154. ……. “Early ART initiation is especially beneficial to pediatric PLWH….. probability of achieving an HIV cure than adult PLWH if receiving early treatment intervention.” Should be expanded and clarified so that readers of the review can clearly understand.

Response: Thank you for your comments. We further explain this point in the revised manuscript at Line 154~161.

Round 2

Reviewer 3 Report

Comments and Suggestions for Authors

The authors addressed convincingly the comments and critiques.